# Point-of-care testing and treatment of sexually transmitted and genital infections during pregnancy in Papua New Guinea (WANTAIM trial): protocol for an economic evaluation alongside a cluster-randomised trial

Neha Batura [1], Olga PM Saweri [2,3], Andrew Vallely [2,3], William Pomat,[3] Caroline Homer,[4,5] Rebecca Guy,[3] Stanley Luchters,[4,6,7,8] Glen Mola,[9] Lisa M Vallely [2,3], Christopher Morgan,[4] Grace Kariwiga,[10] Handan Wand,[2] Stephen Rogerson,[11] Sepehr N Tabrizi,[12] David M Whiley,[13] Nicola Low [14], Rosanna W Peeling,[15] Peter M Siba,[3] Michaela Riddell,[2,16] Moses Laman,[16] John Bolnga,[2,16] Leanne J Robinson,[4,16] Jacob Morewaya,[10] Steven Badman,[2] Angela Kelly-Hanku,[2,3] Pamela J Toliman,[3] Wilfred Peter,[17] Elizabeth Peach,[4] Suzanne Garland,[18] John Kaldor,[2] Virginia Wiseman[2,19]

For numbered affiliations see end of article.

**Correspondence to**
Dr Neha Batura;
n.batura@ucl.ac.uk

## ABSTRACT

**Introduction** Left untreated, sexually transmitted and genital infections (henceforth STIs) in pregnancy can lead to serious adverse outcomes for mother and child. Papua New Guinea (PNG) has among the highest prevalence of curable STIs including syphilis, chlamydia, gonorrhoea, trichomoniasis and bacterial vaginosis, and high neonatal mortality rates. Diagnosis and treatment of these STIs in PNG rely on syndromic management. Advances in STI diagnostics through point-of-care (PoC) testing using GeneXpert technology hold promise for resource-constrained countries such as PNG. This paper describes the planned economic evaluation of a cluster-randomised cross-over trial comparing antenatal PoC testing and immediate treatment of curable STIs with standard antenatal care in two provinces in PNG.

**Methods and analysis** Cost-effectiveness of the PoC intervention compared with standard antenatal care will be assessed prospectively over the trial period (2017–2021) from societal and provider perspectives. Incremental cost-effectiveness ratios will be calculated for the primary health outcome, a composite measure of the proportion of either preterm birth and/or low birth weight; for life years saved; for disability-adjusted life years averted; and for non-health benefits (financial risk protection and improved health equity). Scenario analyses will be conducted to identify scale-up options, and budget impact analysis will be undertaken to understand short-term financial impacts of intervention adoption on the national budget. Deterministic and probabilistic sensitivity analysis will be conducted to account for uncertainty in key model inputs.

**Ethics and dissemination** This study has ethical approval from the Institutional Review Board of the PNG Institute of Medical Research; the Medical Research

## Strengths and limitations of this study

► This protocol will assist in designing economic evaluations for similar complex public health interventions, especially those that seek to capture both health and non-health impacts of point-of-care testing for sexually transmitted infections in low- and middle-income countries (LMICs).

► This protocol follows the Consolidated Health Economic Evaluation Reporting Standards, and guidelines from the Global Health Cost Consortium to design and report economic evaluations nested in a randomised controlled trial and will include individual-level patient cost and health service use data.

Advisory Committee of the PNG National Department of Health; the Human Research Ethics Committee of the University of New South Wales; and the Research Ethics Committee of the London School of Hygiene and Tropical Medicine. Findings will be disseminated through national stakeholder meetings, conferences, peer-reviewed publications and policy briefs.

**Trial registration number** ISRCTN37134032.

## INTRODUCTION

In 2017, it was estimated that every day, globally, more than 1 million people acquire any of the four common curable sexually transmitted infections: chlamydia, gonorrhoea, syphilis and trichomoniasis.[1–3] Left untreated, sexually transmitted and genital infections

such as bacterial vaginosis (henceforth, referred to as STIs) are associated with adverse pregnancy and birth outcomes including spontaneous abortion, miscarriage, stillbirth, pre-term birth, low birth weight, postpartum endometritis, premature rupture of membranes, and various sequelae in newborn infants owing to mother-to-child transmission such as ophthalmia neonatorum.[4–13]

► The planned budget impact and affordability analyses will contribute to the understanding of fiscal space for investments in maternal and child health at provincial and country levels.
► This analysis builds on conventional cost-effectiveness analysis by including non-health benefits such as financial risk protection that are key criteria for equitable resource allocation, and the design of health benefit packages in many LMICs including Papua New Guinea.
► The planned analyses for the base case scenario adopt only a 12-month time horizon but we propose to model the costs and benefits over the lifetime of mother and their babies using published data.

In Papua New Guinea (PNG), prevalence of STIs is high in the general population and among pregnant women.[14] Clinical studies in PNG show that around 50% of all pregnant women test positive for one or more STIs at their first antenatal care (ANC) visit,[6 15] with gonorrhoea, chlamydia, trichomoniasis and bacterial vaginosis most commonly diagnosed.[5 6 15 16] There is evidence from resource-constrained settings to suggest that increased screening for HIV and syphilis in pregnancy is correlated with a reduction in perinatal and infant morbidity and mortality.[17 18] In this high-burden and low-resource setting, poor access to ANC leads to missed opportunities for early diagnosis and clinical intervention.[19]

Traditional STI diagnosis for infections other than HIV and syphilis relies on microscopy, culture, and/or serology that require technical resources and expertise that may not be readily available in all low- and middle-income country (LMIC) settings.[20 21] The long waiting period for results also deters some people from returning to collect their results.[22] In settings where laboratory services are not available, syndromic management, which relies on clinical presentation, is most often used to inform treatment decisions. This strategy fails to accurately identify causative pathogens or detect asymptomatic infections, and consequently leads to negligible impact on health outcomes.[22 23] The development of accurate rapid diagnostic tests for HIV and syphilis used at point-of-care (PoC) has improved their detection, testing coverage and the number of patients accurately diagnosed and treated.[24–26] From an equity standpoint, PoC testing has been shown to improve access to testing and treatment particularly among remote and hard-to-reach populations.[24 27] However, the success of HIV and syphilis PoC diagnosis is yet to be replicated for other common curable STIs, including chlamydia and gonorrhoea.[28–32]

The Women and Newborn Trial of Antenatal Interventions and Management (WANTAIM) study is the first randomised trial to evaluate the effectiveness and cost-effectiveness of PoC STI testing and treatment to improve birth outcomes in high-burden settings.[33] This paper aims to describe the rationale and methodological approach for the economic evaluation of this large-scale trial involving 4600 pregnant women in PNG.

In recent years, the evidence base for the cost and cost-effectiveness of PoC testing for STIs in pregnancy has grown, including in LMICs. A recent systematic review[34] identified that the bulk of these studies was conducted in Africa or Latin and South America,[35–49] with no studies undertaken in East Asia or the Pacific. Most of the studies investigated the cost and cost-effectiveness of PoC testing for syphilis in pregnancy compared with no screening, syndromic management or onsite laboratory testing. Only one study evaluated testing for HIV and testing and treatment for syphilis,[46] another for chlamydia,[39] and none evaluated testing for gonorrhoea, trichomoniasis or bacterial vaginosis. Few studies evaluated the costs and cost-effectiveness of the test and treatment package combined. Despite widespread acknowledgement of the high out-of-pocket costs incurred by women and their families in accessing testing and treatment in many LMICs, the studies in the review were largely conducted from the provider perspective.[35 38–41 43 47–49] Further, none of the studies presented estimates of affordability or budget impact and none analysed non-health-related outcomes such as equity or financial risk protection.[50]

## STUDY SETTING
### Papua New Guinea
In PNG, pregnant women and their infants experience a high burden of adverse health outcomes, with PNG recording one of the highest maternal mortality ratios and neonatal mortality rates in the world: 584 per 100 000 and 25 per 1000 live births, respectively, compared with global figures of 209 and 18.[51 52] In 2012, 20% of births in PNG were preterm birth and/or low birth weight, both key contributors to neonatal mortality.[53]

Pregnant women in PNG experience a high burden of curable STIs. Findings from a country-wide bio-behavioural survey of STIs in pregnancy indicated that the prevalence of chlamydia was 23%, gonorrhoea 14% and trichomoniasis 22%, with 44% of women having at least one of these infections.[5] Another study evaluating the feasibility of a novel PoC testing and treatment strategy for STIs in PNG found that 54% of women had one or more of chlamydia, gonorrhoea, trichomoniasis or bacterial vaginosis, and the prevalence rates of each of these STIs were 19%, 11%, 38% and 18%, respectively.[20] Similar prevalence rates of STIs were observed in a study of malaria prevention in pregnancy.[6] In these studies, between 65% and 80% of infections among pregnant women were asymptomatic indicating the need for more accurate diagnosis at PoC.

In PNG, national guidelines for ANC state that PoC testing and treatment for HIV and syphilis should be

undertaken for all pregnant women at the first ANC clinic visit. For women who test positive, treatment according to national guidelines is offered along with partner testing.[54] However, despite the high prevalence of chlamydia, gonorrhoea, trichomoniasis and bacterial vaginosis among pregnant women, detection and treatment rely on syndromic management according to national guidelines.[54]

### Health services in PNG

In PNG, health services are organised into seven levels of care. Levels 1–4 offer primary care at community aid posts, subhealth centres, health centres, and rural/district hospitals. The majority of level 1–3 facilities are managed and staffed by health extension officers, nursing officers, midwives and community health workers; level 4 facilities, that is, rural/district hospitals usually have a doctor on staff. Population coverage varies from about 5000 to 20 000 per facility and the average distance travelled to reach a facility is 7–8 km. Secondary level care is provided at provincial/regional/national referral hospitals (levels 5–7), which cover an average population of 200 000 and 300 000 in one or more provinces.[55 56] Health workforce distribution is suboptimal, with 0.5 physicians per 10 000 population,[57] compared with the WHO recommended ratio of 10 physicians per 10 000 population.[58] Healthcare is predominantly provided by public health facilities that are either financed and operated by the PNG government or by churches with financial support from the government.[59]

### The WANTAIM trial

WANTAIM aims to test the effectiveness of antenatal PoC testing and treatment for STIs to improve maternal and newborn outcomes in PNG. WANTAIM is being implemented in two provinces in PNG—Madang and East New Britain. Data collection continues despite the challenges of the COVID-19 pandemic, and the trial is due to finish in late 2021.[33]

WANTAIM is a cluster-randomised cross-over trial and the unit of randomisation is a primary healthcare centre and its catchment area. Ten geographically distinct clusters have been assigned in a 1:1 ratio to intervention and control arms in the first phase of the trial. The end of the first phase of the trial is followed by a short washout period of 2–3 months, at the end of which each cluster will cross over to participate in the alternative trial arm in the second phase. The study participants are women attending their first ANC visit, aged over 16 years and less than 26 weeks' gestation (assessed by ultrasound) (n=4600). Newborn infants are followed up within 72 hours of birth.

Pregnant women recruited into the study receive routine ANC as per PNG national guidelines including sulfadoxine/pyrimethamine for malaria prevention; iron and folate supplementation; tetanus toxoid immunisation; HIV and syphilis screening (and treatment if required). Women in the control arm receive STI syndromic management if they report symptoms of a genital infection (abdominal pain, discharge). They also provide a urine sample for diagnostic testing on GeneXpert in the study laboratory. If positive for STI at their last test, they receive treatment during the first postnatal visit. Women in the intervention arm of the trial provide a self-collected vaginal specimen for PoC STI testing, and same-day treatment as necessary, at the following time points:

► At enrolment (<26 weeks' gestation).
► One month after trial enrolment.
► At 34–36 weeks' antenatal follow-up.

The primary outcome of the trial is a composite measure of two events, the proportion of women and their newborn infants in each trial arm who experience either a preterm birth (<37 weeks' gestation) and/or low birth weight (<2500 g).

The study has ethical approval from the Institutional Review Board (IRB) of the PNG Institute of Medical Research (IRB number 1608); the Medical Research Advisory Committee (MRAC) of the PNG National Department of Health (MRAC number 16.24); the Human Research Ethics Committee (HREC) of the University of New South Wales (HREC number 16708); and the Research Ethics Committee (REC) of the London School of Hygiene and Tropical Medicine (REC number 12009).

A full description of the WANTAIM intervention and trial design is described elsewhere.[33] The purpose of this paper is to fully describe the methods for the economic evaluation of the trial.

## AIMS AND OBJECTIVES

The economic evaluation aims to assess the cost-effectiveness and affordability of PoC testing and treatment of curable STIs in pregnancy compared with standard care from a provider and societal perspective. The specific objectives of the economic evaluation are to:

1. Estimate total financial and economic costs of the PoC STI intervention.
2. Model incremental cost-effectiveness of the intervention compared with standard care.
3. Extend the incremental cost-effectiveness analysis (CEA) to include equity-related measures of impact.
4. Conduct a budget impact analysis to assess the affordability of implementing the intervention at the national level or in target areas/populations.

These planned analyses will adhere to the Consolidated Health Economic Evaluation Reporting Standards and established guidelines from the Global Health Cost Consortium for conducting and reporting economic evaluation for global health trials.[60 61]

## METHODS: ECONOMIC EVALUATION OF THE WANTAIM TRIAL

### Costing data

Cost data collection is guided by the perspective adopted for the economic evaluation. For WANTAIM, direct and

**Table 1** Cost category and data sources

| Description | Type of cost | Data sources | Sample size |
|---|---|---|---|
| **Provider costs** | | | |
| Costs of implementing WANTAIM | Direct | Project accounts of implementing agencies | N/A |
| Cost of providing ANC services | Direct | Health facilities | 10 health facilities |
| Cost of increased workload of facility staff associated with PoC testing and treatment | Indirect | Patient pathway and health worker observation data collected as part of the health facility assessment | 20–30 |
| **Participant costs** | | | |
| Costs of care-seeking | Direct and indirect | Participant case report forms | 4600 |

ANC, antenatal care; N/A, not applicable; PoC, point-of-care; WANTAIM, Women and Newborn Trial of Antenatal Interventions and Management.

indirect costs will be collected from the provider perspective and societal perspective—the latter including any costs incurred by pregnant women and their families. The different cost categories and data sources are summarised in table 1. A combination of top-down and bottom-up costing approaches will be used[60] and the time horizon for the main trial-based economic evaluation will be 12 months.

*Provider costs* are incurred by the institutions implementing the PoC testing intervention across the start-up, implementation and monitoring phases of the trial. The cost data will be sourced from financial records, programme documents and consultation with project staff. A step-down costing methodology will be used, whereby costs from project accounts are entered into a customised tool created in Microsoft Excel, which is adapted each year to reflect the changing cost structure of the trial during the start-up and implementation phases.

Financial costs will be converted to economic costs, that is, any donated goods or volunteer time that do not appear in the programme accounting data will be added to the cost sheets and assigned a current market value.[62 63] Key informant interviews with programme leads will assist in identifying donated or subsidised items and in allocating joint costs between programme components. The allocation of joint staff costs will be informed by monthly staff timesheets. Research costs will not be included in the CEA. However, start-up costs will be reported and differentiated from implementation costs to enable decision-makers to gauge the costs associated with the initial activities and expenditures necessary to develop PoC testing and integration with standard ANC.[64]

*Provider (treatment) costs* are incurred by provincial health authorities, who manage ANC, delivery and postnatal visits; and church health services, non-state providers who access a mix of government and institutional funds. Primary data on the average unit cost of care will be collected from all health facilities participating in the WANTAIM trial. A simple cost-capture form has been developed for facility data collection adapted from other costing studies led by members of this team.[65 66] Data from this form will be used to complement existing data from centre reports, patients' records and published national reports relating to ANC, labour and birth care, and postnatal care visits. Costs of services provided will also be calculated using a step-down approach.[67]

*Participant (treatment) costs* are the direct and indirect costs of healthcare seeking incurred by women and their families such as medical costs, transport costs and the opportunity costs in terms of lost productivity due to care-seeking visits. These will be estimated for standard treatment episodes in the control arms and for treatment episodes in the intervention arm to gauge changes in out-of-pocket costs of care-seeking and time dedicated to care-seeking for participants. Data on the direct and indirect costs incurred by participants are being collected from all trial participants (n=4600) in both arms of the trial at enrolment and three follow-up visits through participant case report forms (CRFs). The participant cost data will be summed and analysed as cross-sectional data to gauge the economic burden borne by participants and their households that is alleviated due to PoC testing and treatment of STIs in pregnancy.

### Health service use
Health service utilisation for all trial participants in the intervention and control arms will be estimated using data collected via a take-home aide memoire and participant CRFs. The aide memoire is provided to all participants at recruitment, who use this tool to make notes about the facility visits that they make or attend between the WANTAIM follow-up visits. The aide memoire also allows them to make notes about any costs associated with those visits. At the WANTAIM follow-up visits, these notes serve as prompts for questions about service utilisation and costs of care-seeking, which are recorded in the CRFs.

### Proposed analyses
#### Cost and CEAs
A base case analysis will be undertaken alongside the trial to estimate the cost-effectiveness of the intervention compared with standard care as implemented. The base case will include all start-up costs and implementation costs. Costs will be presented in current prices in PNG kina and international dollars (INT$). All costs will be adjusted for inflation using the Consumer Price Index for PNG and will be converted to 2021 INT$ using the 2021 Purchasing Power Parity conversion factor for PNG. Costs and outcomes will be converted to present values using an annual discount rate of 3% in the base case, and annual rates of 0%, 6% and 9% in sensitivity analyses.

For the base case analysis, results will be presented in terms of total financial and economic costs of the

intervention and incremental cost-effectiveness ratios (ICERs) for the primary outcome, that is, the proportion of women and their newborns who experience either preterm birth and/or low birth weight. ICERs will be calculated as the arithmetic mean difference in cost between the intervention and control arms, divided by the arithmetic mean difference in effect. To maximise comparability with other trials, ICERs will also be reported in terms of cost per life year saved and cost per disability-adjusted life year (DALY) averted (see the Modelling section for details).

A descriptive analysis of missing data will be undertaken to inform the base case assumption regarding the missing data mechanism (the probability that missing data are independent or not on the observed or unobserved values). Appropriate methods will be used to handle the missing data, which may include mean imputation, multiple imputations, available case analysis, inverse probability weighting or likelihood methods.[68] Sensitivity analyses will be conducted as appropriate.

The data on costs and outcomes for the period of trial follow-up will be at the individual level, allowing evaluation of uncertainty of the cost-effectiveness estimates using non-parametric bootstrapping.[69] Cost-effectiveness acceptability curves (CACs) will be generated to further describe uncertainty around the cost estimates.[70] CACs indicate the proportion of the estimates produced by bootstrapping that would be 'acceptable' below a range of willingness-to-pay thresholds, where willingness to pay is the value placed on an additional pregnant woman appropriately tested for STIs in pregnancy. Sensitivity analyses will take into account the uncertainty in key parameters that may have been affected by the COVID-19 pandemic, such as staff or drug costs.

### Modelling

The base case analysis will have a time horizon of up to 12 months. If the intervention demonstrates clinical effectiveness over that period, we will employ a cohort decision analytical model to examine the cost-effectiveness of the intervention over a newborn's lifetime. A Markov model will be used to estimate the long-term health benefits, healthcare costs and cost-effectiveness of the PoC intervention compared with standard care, drawing on results of the WANTAIM trial and available published data. The point of entry into the model will be 'tested for STIs'. There are two possible states for women: infected or uninfected. Women identified as infected and then treated may recover and stay healthy, become re-infected or die. Health outcomes will therefore depend on treatment compliance and include live birth without infection (healthy infant), live birth with infection, preterm and/or low birth weight, and neonatal death. The model will be used to project differences between the intervention and control arms in life years saved, DALYs averted and lifetime healthcare costs. Sensitivity analyses will also be conducted within this model.

### Equity impact of the intervention

The equity impact of the intervention will be investigated by conducting an extended CEA (E-CEA). The E-CEA broadens the scope of the CEA by incorporating health equity and financial protection considerations for the most vulnerable sections of the population that are likely to have the highest need.[71] This will be done across three domains: by exploring improving health gains, with particular reference to the poorest socioeconomic group; reduction in the out-of-pocket expenses faced by households seeking care; and improved financial protection or reducing the number of households that sink into poverty due to catastrophic health spending.[72] For the E-CEA, provider and participant cost data (table 1) will be synthesised with data on service utilisation that will be collected via participant CRFs that are completed at enrolment into the trial and at three follow-up trial visits. All results will be presented by socioeconomic quintiles. Given that socioeconomic groups may not differ greatly within clusters, a Multidimensional Poverty Index (MDPI) will be derived from socioeconomic and income data collected from all trial participants at enrolment. The use of an MDPI provides a more nuanced understanding of socioeconomic status of households as it takes monetary and non-monetary dimensions of deprivation into account.[73] This enables the differentiation between population groups who may all be relatively poor in monetary dimensions such as income or asset ownership.[74] Thus, the consideration of other non-monetary attributes (eg, housing) allows us to distinguish between households that are homogeneously asset or cash poor in this study setting.[74]

### Scale-up and budget impact analysis

The costs and cost-effectiveness of the intervention will also be considered in a scale-up scenario, in which any start-up costs will be excluded as they are considered sunk costs.[75] The budget impact of the scale-up scenario will be explored by an analysis of fiscal space for programme delivery using a generalised fiscal space assessment method[76 77] and probabilistic analyses to determine a set of cost-effectiveness thresholds.[70 78]

### Patient and public involvement

WANTAIM trial participants were involved in providing data for the study.

### DISCUSSION

To our knowledge, this paper is the first protocol for the economic evaluation of PoC STI testing and treatment in pregnancy in an LMIC setting. The proposed analyses aim to assess the cost-effectiveness of the intervention as well as its affordability and equity impact. The analyses will adhere to international guidelines for conducting and reporting economic evaluation studies and provide transparency in how they are conducted. The findings of the economic evaluation will provide decision-makers in

PNG and similar settings evidence on the relative value for money of this intervention and the likely level of investment required for implementation at scale. The findings of this study will be disseminated through national stakeholder meetings, conferences, peer-reviewed publications and policy briefs.

## Dissemination

The findings of the economic evaluation of the WANTAIM trial will be disseminated to academic and policymaking communities, and the wider public, in peer-reviewed journals, and presented at relevant conferences in PNG and globally.

**Author affiliations**
[1]Institute for Global Health, University College London, London, UK
[2]The Kirby Institute, University of New South Wales, Sydney, New South Wales, Australia
[3]The Papua New Guinea Institute of Medical Research, Goroka, Papua New Guinea
[4]The Burnet Institute, Melbourne, Victoria, Australia
[5]Faculty of Health, University of Technology Sydney, Sydney, New South Wales, Australia
[6]Department of Public Health and Preventive Medicine, Ghent University, Ghent, Belgium
[7]Department of Population Health, Medical College, Aga Khan University, Nairobi, Kenya
[8]Department of Epidemiology and Preventive Medicine, Monash University, Monash, Victoria, Australia
[9]School of Medicine and Health Sciences, University of Papua New Guinea, Port Moresby, Papua New Guinea
[10]Milne Bay Provincial Health Authority, Alotau, Papua New Guinea
[11]Department of Medicine, The Doherty Institute, University of Melbourne, Melbourne, Victoria, Australia
[12]The Royal Women's Hospital, Parkville, Victoria, Australia
[13]The University of Queensland Centre for Clinical Research, Faculty of Medicine, The University of Queensland, Brisbane, Queensland, Australia
[14]Institute of Social and Preventive Medicine, University of Bern, Bern, Switzerland
[15]Department of Clinical Research, London School of Hygiene and Tropical Medicine, London, UK
[16]Papua New Guinea Institute of Medical Research, Madang, Papua New Guinea
[17]Madang Provincial Health Authority, Madang, Papua New Guinea
[18]Microbiology and Infectious Diseases Department, The Royal Women's Hospital, Parkville, Victoria, Australia
[19]Department of Global Health and Development, London School of Hygiene and Tropical Medicine, London, UK

**Acknowledgements** The study team would like to thank the WANTAIM trial participants, health facility staff and the provincial officers in PNG.

**Contributors** VW, NB and OPMS led the design of the economic evaluation. NB, OPMS, VW, AV, WP, CM and MR contributed to data collection and acquisition for the economic evaluation. NB, OPMS and VW will lead the formal data analysis for the economic evaluation. AV, WP, CH, RG, SL, GM, LV, CM, GK, HW, SR, SNT, DMW, NL, RWP, PMS, MR, ML, JB, LR, JM, SB, AK-H, PJT, WP, EP, SG, JK and VW are members of the research team involved in conceptualisation, funding acquisition, methodology, data curation and/or formal analysis of the WANTAIM trial within which this economic evaluation is nested, and from where the outcome data for the economic evaluation will be used. NB was responsible for the initial drafting of this manuscript. All authors contributed substantially to the review of this manuscript and provided critical and constructive comments. All authors read and approved the final manuscript. NB, OPMS, VW, AV, LV, MR and SB contributed to responding to reviewer comments and the revision of the manuscript.

**Funding** This study is funded by the Joint Global Health Trials scheme (UK Department for International Development, Medical Research Council, Wellcome Trust; MR/N006089/1); National Health and Medical Research Council, Australia

(Project Grant 1084429); and Swiss National Science Foundation Research for Development award (IZ07Z0_160909/1).

**Competing interests** None declared.

**Patient and public involvement** Patients and/or the public were involved in the design, or conduct, or reporting, or dissemination plans of this research. Refer to the Methods section for further details.

**Patient consent for publication** Not required.

**Provenance and peer review** Not commissioned; externally peer reviewed.

**ORCID iDs**
Neha Batura http://orcid.org/0000-0002-8175-8125
Olga PM Saweri http://orcid.org/0000-0001-5142-8355
Andrew Vallely http://orcid.org/0000-0003-1558-4822
Lisa M Vallely http://orcid.org/0000-0002-8247-7683
Nicola Low http://orcid.org/0000-0003-4817-8986

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
