## [Reviewer comments · BMJ Open]

ARTICLE DETAILS

TITLE (PROVISIONAL)	Protocol for an economic evaluation alongside a cluster-randomised trial of point-of-care testing and treatment of sexually transmitted and genital infections during pregnancy in Papua New Guinea (WANTAIM trial)
AUTHORS	Batura, Neha; saweri, olga; Vallely, Andrew; Pomat, Willie; Homer, Caroline; Guy, Rebecca; Luchters, Stanley; Mola, Glen; Vallely, Lisa; Morgan, Christopher; Kariwiga, Grace; Wand, Handan; Rogerson, Stephen; Tabrizi, Sepehr; Whiley, David; Low, Nicola; Peeling, Rosanna; Siba, Peter; Riddell, Michaela; Laman, M.; Bolnga, John; Robinson, Leanne; Morewaya, Jacob; Badman, Steven; Kelly-Hanku, Angela; Toliman, Pamela; Peter, Wilfred; Peach, Elizabeth; Garland, Suzanne; Kaldor, John; Wiseman, Virginia

VERSION 1 – REVIEW

REVIEWER	Manoukian, Sarkis Glasgow Caledonian University, School of Health and Life Sciences
REVIEW RETURNED	23-Dec-2020

GENERAL COMMENTS	This is a protocol of a study with important implications for health service delivery in PNG. This was a clearly written well-described protocol and I would like to congratulate the authors for that and wish them good luck for the completion of the trial and statistical/economic analysis. I have three questions for the authors and I would have liked to see these addressed in the main text before this protocol is published: 1) I do not see anything about missing data and imputation. This tends to be an important aspect in the statistical analysis of such trials. Can you explain what is the expectation for missing data, how these are going to be dealt with and describe appropriate multiple imputation methods. I think the best way is to do this when you analyse cost-effectiveness with uncertainty i.e. not in the base case analysis but treat this as a suggestion, feel free to explain a different approach. 2) Good to see a decision analytic model and this will make your analysis stronger. I would have liked to have seen a bit more detail about the model and type of modelling analysis e.g. cohort versus patient level or perhaps an early model diagram with Markov
---

	states. I don't expect to see all the details at this stage but something more than a single sentence would be welcome. 3) I see that you have follow-ups but it is not clear how the analysis will account for longitudinal nature of the data. Can you explain this? Is this only about summing up all the costs and treating the data as cross-sectional?
--	--

REVIEWER	Peters, Remco Foundation for Professional Development, Research Unit
REVIEW RETURNED	04-Jan-2021

GENERAL COMMENTS	Thank you for the invitation to review this manuscript. I would like to congratulate Dr Batura and Team for a great research protocol and look forward to the results of their proposed economic evaluation. Undoubtedly, these will provide a significant contribution to improving the health and wellbeing of newborns in LMICs. I only have two comment/question: 1) Are the authors planning to take into account any women presenting with STI-associated symptoms at time points other than the three moments of STI screening? And women with persistent/recurrent symptoms? 2) Considering the high prevalence: are the authors planning to include presumptive treatment for all pregnant women as potential alternative scenario to diagnostic testing?
--

VERSION 1 – AUTHOR RESPONSE

Reviewer 1:

This is a protocol of a study with important implications for health service delivery in PNG. This was a clearly written well-described protocol and I would like to congratulate the authors for that and wish them good luck for the completion of the trial and statistical/economic analysis.

Response: *Thank you very much for your positive feedback and wishes.*

I have three questions for the authors and I would have liked to see these addressed in the main text before this protocol is published:

1) I do not see anything about missing data and imputation. This tends to be an important aspect in the statistical analysis of such trials. Can you explain what is the expectation for missing data, how these are going to be dealt with and describe appropriate multiple imputation methods. I think the best way is to do this when you analyse cost-effectiveness with uncertainty i.e. not in the base case analysis but treat this as a suggestion, feel free to explain a different approach.

Response: *A descriptive analysis of the missing data will be undertaken to inform the base-case assumption regarding the missing data mechanism (the probability that missing data are independent or not on the observed or unobserved values). Appropriate methods will be used to handle the missing data, which may include mean imputation, multiple imputation, available case analysis, inverse probability weighting, or likelihood methods, will be selected. Sensitivity analysis will be conducted as appropriate. The text on page 18 (lines 3-12) has been revised to include this.*

2) Good to see a decision analytic model and this will make your analysis stronger. I would have liked to

have seen a bit more detail about the model and type of modelling analysis e.g. cohort versus patient level or perhaps an early model diagram with Markov states. I don't expect to see all the details at this stage but something more than a single sentence would be welcome.

Response: Thank you for this suggested. We have provided further detail in the 'Modelling' sub section on page 18 (lines 33-54).

3) I see that you have follow-ups but it is not clear how the analysis will account for longitudinal nature of the data. Can you explain this? Is this only about summing up all the costs and treating the data as cross-sectional?

Response: The costs incurred by participants and their households at all follow-up visits, as well as interim health care seeking visits will be summed to estimate the total costs of care seeking during pregnancy, delivery and in the post-natal period. This will allow us to gauge the economic burden borne by participants and their households that is alleviated due to point of care (PoC) testing and treatment of STIs in pregnancy. This has been clarified on page 17 (lines 7-10)

Reviewer: 2

Thank you for the invitation to review this manuscript. I would like to congratulate Dr Batura and Team for a great research protocol and look forward to the results of their proposed economic evaluation. Undoubtedly, these will provide a significant contribution to improving the health and wellbeing of newborns in LMICs.

Response: Thank you very much for your positive feedback

I only have two comment/question:

1) Are the authors planning to take into account any women presenting with STI-associated symptoms at time points other than the three moments of STI screening? And women with persistent/recurrent symptoms?

Response: The WANTAIM trial will collect data on women returning at any of the scheduled three visits after the first enrolment or any unscheduled visits. Women will be treated if they present with symptoms suggesting an STI. The trial will also use reports of serious adverse events to capture any women who present to emergency or treatment clinics and treated for STI during pregnancy (symptomatic). Thus, this will be taken into account.

2) Considering the high prevalence: are the authors planning to include presumptive treatment for all pregnant women as potential alternative scenario to diagnostic testing?

Response: The aim of the WANTAIM trial is to evaluate the effectiveness and cost effectiveness of PoC testing and treatment of STIs in pregnancy, compared to standard treatment i.e. syndromic management in Papua New Guinea (PNG). Thus, data on presumptive treatment is not being collected within the trial, and as such is outside the scope of its economic evaluation. There are several challenges to considering presumptive treatment for all pregnant women as an alternative scenario. First, it is challenging to understand when women might receive treatment, for example, could this be at every antenatal care (ANC) visit? This may not be feasible as some women may have as many as 8-9 ANC visits, while some women have one or none. Second, the presumptive treatment of partners would also need to be included. Third, as these data are not collected within the trial, they would have to be derived from another context. Given the unique context of PNG in terms of disease burden, service utilization patterns, and health system organisation, we do not believe that this is a valid option.

VERSION 2 – REVIEW

REVIEWER	Manoukian, Sarkis Glasgow Caledonian University, School of Health and Life Sciences
REVIEW RETURNED	04-May-2021

GENERAL COMMENTS	In my opinion this paper is acceptable for publication
--